

**Spatial and seasonal variations of leaf area index (LAI) in subtropical secondary**
**forests related to floristic composition and stand characters**
**Wenjuan Zhu[1,2], Wenhua Xiang[1,2,3*], Qiong Pan[4,1], Yelin Zeng[1], Shuai Ouyang[1,2,3], Pifeng Lei[1,2,3],**
**Xiangwen Deng[1,2,3], Xi Fang [1,2,3], Changhui Peng[5,1]**
Faculty of Life Science and Technology, Central South University of Forestry and Technology,
Changsha 410004, Hunan Province, China
Huitong National Field Station for Scientific Observation and Research of Chinese Fir Plantation
Ecosystem in Hunan Province, Huitong 438107, China
National Engineering Laboratory of Applied Technology for Forestry & Ecology in Southern China,
Changsha 410004, China
Changsha Environmental Protection College, Changsha 410004, China
Institute of Environment Sciences, Department of Biological Sciences, University of Quebec at
Montreal, Montreal, QCH3C 3P8, Canada
*Correspondence to*: Wenhua Xiang, Email: xiangwh2005@163.com, Tel.: +86 0731 85623483





**Abstract.** Leaf area index (LAI) is an important parameter related to carbon, water and energy
exchange between canopy and atmosphere, and is widely applied in the process models to simulate
production and hydrological cycle in forest ecosystems. However, fine-scale spatial heterogeneity of
LAI and its controlling factors have not been fully understood in Chinese subtropical forests. We used
hemispherical photography to measure LAI values in three subtropical forests (i.e. *Pinus massoniana -*
*Lithocarpus glaber* coniferous and evergreen broadleaved mixed forests, *Choerospondias axillaris*
deciduous broadleaved forests, and *L. glaber - Cyclobalanopsis glauca* evergreen broadleaved forests)
during period from April, 2014 to January, 2015. Spatial heterogeneity of LAI and its controlling factors
were analysed by using geostatistics method the generalised additive models (GAMs), respectively. Our
results showed that LAI values differed greatly in the three forests and their seasonal variations were
consistent with plant phenology. LAI values exhibited strong spatial autocorrelation for three forests
measured in January and for the *L. glaber - C. glauca* forest in April, July and October. Obvious patch
distribution pattern of LAI values occurred in three forests during the non-growing period and this
pattern gradually dwindled in the growing season. Stand basal area, crown coverage, crown width,
proportion of deciduous species on basal area basis and forest types affected the spatial variations in
LAI values in January, while species richness, crown coverage, stem number and forest types affected
the spatial variations in LAI values in July. Floristic composition, spatial heterogeneity and seasonal
variations should be considered for sampling strategy in indirect LAI measurement and application of
LAI to simulate functional processes in subtropical forests.




**Keywords:** Leaf area index; Spatial heterogeneity; Deciduous species; Generalised additive models (GAMs)

# 1 Introduction

Many fundamental ecological processes in forest ecosystems, such as carbon (C) flux as well as water and energy exchanges, take place between canopy layer and the atmosphere (Brut et al., 2009; GCOS, 2006; Alonzo et al., 2015; Liu et al., 2015b). At finer scale, leaves within canopy are the primary organ to perform a series of physiological activities (i.e. photosynthesis, respiration and evapotranspiration) (Aragão et al., 2005) and physical reactions (i.e. rainfall and radiation interception) (Smith, 1981; Crockford & Richardson, 2000; Aston, 1979). Therefore, the amount of leaves in a forest is the determinant of aboveground ecological processes and ecosystem functions. Leaf area index (LAI), defined as total one-sided leaf area per unit ground surface area (Biudes et al., 2014), is widely used parameter (Kross et al., 2015) to quantitatively describe the vegetation canopy structure (Woodgate et al., 2015), to simulate ecological process models (Sprintsin et al., 2007; Facchi et al., 2010; Brooks et al., 2006; Gonsamo & Chen, 2014) and to reveal tree growth and productivity in forests at stand scale and landscape level (Liu et al., 2015b; Lee et al., 2004). In addition, LAI is listed as one of the essential variables for observation of global climate (Mason et al., 2003; Manninen et al., 2009) and for remote sensing data validation (Asner et al., 2003; Clark et al., 2008). Thus, accurate estimations of LAI value are important to understand ecological processes in forest ecosystems.



At present, various direct and indirect methods have been developed to measure LAI in forests.
Direct estimation methods including leaf harvest (Clark et al., 2008), allometric equations and litter
collection (Ryu et al., 2010; Liu et al., 2015a) are recognised as the most accurate method. However,
leaf harvest and allometric equations methods need time-consuming, labour-intensive and destructive
sampling process, while litter collection is more feasible for temperate deciduous forests. Obviously, the
direct method is less applicable for large-scale and long-term LAI monitoring (Bequet et al., 2012;
Biudes et al., 2014). Indirect methods include plant canopy analyser (Licor LAI-2000), hemispherical or
fisheye photography (Macfarlane et al., 2007) and remote sensing (Biudes et al., 2014). The indirect
methods retrieve LAI value from light transmittance through canopies or from canopy image analysis.
For large scale LAI estimates, remote sensing is the effective method but requires validation with the
ground-based LAI data. It is still a challenge for LAI estimates on the ground at small scales due to the
problems of sampling strategies associated with accepted level of accuracy, time and cost consideration
(Richardson et al., 2009). Hemispherical photography is a relatively simple and easily operated method
among many indirect methods to retrieve LAI value at small scales (Demarez et al., 2008). Correction
of the effects of woody materials, clumping and zenith angels or exposure is critical for improving the
accuracy of LAI estimation (Liu et al., 2015b). Analysis software development and portable and timely
characteristics allow hemispherical photography to measure spatial heterogeneity and seasonal
variations of LAI in forests.
Forest canopy structure is highly complicate so LAI values show great temporal and spatial variations



at scales ranging from stand to global scale. For example, LAI values in the 7.9 ha plot of an old humid
temperate forest tended to increase spatially as elevation increased and showed a temporal variation
with plant phenology (Naithani et al., 2013). The spatial patterns of LAI values at stand scale were
significantly influenced by spatial distribution of tree species, which was dependent on topography and
soil types (Naithani et al., 2013). Coefficient of variations (CV) in LAI decreased as the scale increased
and LAI values did not have any relationship with biome type and climate patterns, but were influenced
by land use and land cover, terrain features, and soil properties at stand scale (Aragão et al., 2005). The
CV of LAI of three species (i.e. beech, oak and pine) had different degree spatial variations in 1 ha plot
at stand level (Bequet et al., 2012). LAI values in sagebrush displayed strong spatial patterns with the
time after disturbance and increased with stand age and total plant cover (Ewers & Pendall, 2007). The
LAI values derived from MODIS data (Huang et al., 2008; Myneni et al., 2002) revealed strong spatial
variations at global scale and the variations were correlated with latitude (Tian et al., 2004). At global
scale, temperature is the limiting factor for LAI under cool conditions while water plays a predominant
role under other conditions, and this pattern differed among plant functional types (Iio et al., 2014). The
factors that govern the spatial variations in LAI values at stand level include forest types, stand structure
(Bequet et al., 2012), climate (Shao & Zeng, 2011), topography, soil moisture condition (Breshears &
Barnes, 1999), and human disturbance and management activities (Huang & Ji, 2010). Although effects
of topography, soil properties (Naithani et al., 2013; Aragão et al., 2005) and stand character (Bequet et
al., 2012; Yao et al., 2015) on LAI values have been investigated in detail, how forest types, tree species



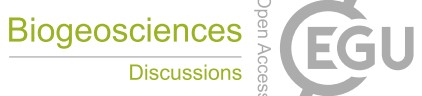

diversity and stand structure affect spatial heterogeneity and seasonal variations of LAI has not been
fully understood.
Chinese subtropical forests contain a diversity of tree species with complex canopy structure and
mostly grow on heterogeneous topography and soil condition. As a result, LAI in subtropical forests
may exhibit greatly spatial and seasonal variations, which is worthy of further investigation. However,
LAI data of subtropical forests are relative deficiency in the global database (see Asner et al., 2003). In
this study, we selected three different forests (i.e. *Pinus massoniana - Lithocarpus glaber* coniferous
and evergreen broadleaved mixed forests, *Choerospondias axillaris* deciduous broadleaved forests, and
*L. glaber - Cyclobalanopsis glauca* evergreen broadleaved forests), LAI values were measured by using
hemispherical photography. Spatial heterogeneity of LAI was investigated through geostatistics analysis
method. The generalised additive models (GAMs) were used to examine how tree species diversity and
stand characters affect LAI variations in the three forests. Specifically, the objectives of this study were:
(1) to examine differences and seasonal variations in LAI among three forests in subtropical China; (2)
to analyse spatial heterogeneity of LAI values within a specific forest; and (3) to identify how forest
types, species diversity and stand characters control the spatial heterogeneity and seasonal variations of
LAI values in three forests.

## 113   2 Materials and methods

### 114   2.1 Study site description



The study was carried out at the Dashanchong Forest Farm (latitude 28°23′58″ - 28°24′ 58″ N,
longitude 113°17′46″ - 113°19′08″ E), Changsha County, Hunan Province, China. The Farm
experiences a humid mid-subtropical monsoon climate. Mean annual air temperature was between
16.6 ℃ to 17.6 ℃, with a mean monthly minimum temperature of -11 ℃ in January and maximum
temperature of 40 ℃ in July. Mean annual precipitation ranged from 1412 mm to 1559 mm, mostly
occurring between April and August. The topography is characterized by a typical low hilly landscape
with an altitude between 55 m to 260 m above sea level. Soil type is designated as well-drained clay
loam red soil developed on slate and shale rock, classified as Alliti-Udic Ferrosols, corresponding to
Acrisol in the World Reference Base for Soil Resource (IUSS Working Group WRB, 2006). Evergreen
broadleaved forest is the climax vegetation of the region. As a result of human disturbance and
management activities, the Farm possesses a range of secondary forests dominated by different tree
species, including *P. massoniana - L. glaber* coniferous and evergreen broadleaved mixed forests, *C.*
*axillaris* deciduous broadleaved forests, and *L. glaber - C. glauca* evergreen broadleaved forests (Xiang
et al., 2015).

**2.2 Stand characteristics determination**
We established a permanent plot for each of three forests (i.e. 90 m × 190 m plot for *P. massoniana -*
*L. glaber* mixed forests, 100 m × 100 m plot for *C. axillaris* deciduous forests, and 100 m × 100 m plot
for *L. glaber - C. glauca* evergreen broadleaved forests). Each plot was divided into 10 m × 10 m



subplots, where tree species, diameter at breast height (DBH, cm), tree height (H, m), height under the
lowest live branch (m) and crown width (m) were measured for the individual stem with DBH larger
than 1 cm. Stand characteristics for the trees with DBH larger than 4 cm of the three forests were
presented in Table S1.

To identify the factors that control spatial heterogeneity of LAI values in the forests, we selected

individual trees with H larger than average height of each stand (see Table S1) and calculated their stem
number, average DBH, H, total basal area at breast height (BA), crown width, crown coverage,
biodiversity index, the proportion of BA of three functional group (coniferous, deciduous and evergreen
broadleaved species) to total stand BA within a subplot. Biodiversity index (BDI) was determined by
Shannon-Wiener index as the formula:

$BDI = -\sum P_i \ln P_i$ (1)

where $P_i$ is important value of $i$th species and is calculated by dividing the sum of relative abundance
degree ($A$r) and relative dominance degree ($D$r) of $i$th species within a subplot by two.

**2.3 Sampling design for LAI measurement**

At the centre of each subplot of the three forests, hemipherical photographs were taken using a LAI

measuring instrument (SY-S01A) throughout four measurement seasons, i.e. in April (spring), July
(summer) and October (autumn) in 2014 and January (winter) in 2015. The operation was carried out
below canopy with the fisheye lens 1.0 m above the ground (Manninen et al., 2009). We took the



photographs in the morning, dusk or cloudy to minimize influence of direct sunshine (Rich, 1990;
Bequet et al., 2012). The images were processed and effective LAI values ($L_e$) were recorded by the
plant canopy analysis system of the LAI measuring instrument. To obtain accurate LAI ($L$), the
correction was made to $L_e$ based on previous theory (Chen, 1996):
$$L = \frac{(1-\alpha)L_e\gamma_E}{\Omega_E}$$
(2)

where $\alpha$ is the ratio of woody to total area and reflects the contribution of woody materials to $L_e$. $\Omega_E$ is
the clumping index that quantifies the effect of foliage clumping beyond shoots level. $\gamma_E$ is the needle to
shoot area ratio and quantifies the effect of foliage clumping within shoots.
Photoshop Software (Adobe Photoshop CS5, Adobe Systems Incorporated, North America) was used
to calculate $\alpha$. After total pixel number of $L_e$ image was determined, the Clone Stamp Tool in the
software was used to replace the woody materials with surrounding of non-woody materials and to
obtain the pixel number and recorded as LAI of leaves ($LAI_{leaf}$). The value of $\alpha$ was calculated
accordingly:
$$\alpha = (L_e - LAI_{leaf})/L_e$$
(3)

The logarithm averaging method proposed by Lang and Xiang (1986) was applied to calculate $\Omega_E$:
$$\Omega(\theta) = \frac{\ln[P(\bar{\theta})]}{\ln[\overline{P(\theta)}]} = \frac{n\ln[P(\bar{\theta})]}{\sum_{k=1}^{n}\ln(P_k(\theta))}$$
(4)

where $P(\theta)$ is the average gap fraction (expressed without the bar in the text), $\ln[P(\theta)]$ is the logarithm
average of the gap fraction, $P_k(\theta)$ is the gap fraction of segment $k$. For deciduous and evergreen



broadleaved species, $\gamma_E$=1.0, but for coniferous species, $\gamma_E$ is always greater than 1.0, but we ignored the
effect of needle to shoot area on LAI in this study.

**2.4 Data analysis**
The minimum, maximum, mean value, standard deviation and coefficient of variation (CV) were
calculated for the LAI data in the three forests. Two-way analysis of variance (ANOVA) was used to
detect effects of forest types and measurement seasons on LAI values. The LAI data in the three forests
were tested for normal distribution using the K-S test ($p<0.05$). We used domain method to identify the
specific values and replaced them with normally maximal values and if the data did not meet normal
distribution, the transformation was applied till the statistical assumption was met. Most values required
natural logarithm transformation to meet assumptions of normality. The exception is for *L. glaber - C.*
*glauca* in April which was cosine transformed and *L. glaber - C. glauca* in November which was
artan-transformed.
To investigate spatial heterogeneity of LAI values at four seasons measured in the three forests,
semivariance function was calculated as the following formula:
$$\gamma(h) = \frac{1}{2N(h)} \sum_{i=1}^{N(h)} [Z(x_i) - Z(x_i + h)]^2 \tag{5}$$

where $\gamma(h)$ is semivariance value of lag distance $h$, $N(h)$ is the number of pair data for lag distance $h$,
$Z(x_i)$ and $Z(x_i+ h)$ represent LAI values at coordinate $x_i$ and $(x_i+h)$ (Rossi et al., 1992). Based on the
semivariogram plotting $\gamma(h)$ values against $h$ variable, the appropriate models were fitted and we





obtained the values of nugget ($C_0$), sill ($C_0+C$), range ($A_0$) (Ewers & Pendall, 2007) and the ratio
$[C/(C_0+C)]$ that reflected the degree of spatial autocorrelation of LAI values in a forest. Because spatial
autocorrelation and semivariogram theory make unbiased optimal estimation for regional variables in a
limited area (Bivand et al., 2013), Kriging interpolation method was used to predict unknown LAI
values in the forests from the data measured and to produce spatial distribution map of LAI values for
the three forests at four seasons.

Because the largest amount of defoliated leaves occurs in January and leaves fully expand in July in

subtropical forests, we chose LAI values measured in January and July in three forests as response
variable. Stepwise regression was used to select the factors that significantly affect LAI variations. The
factors include forest types, species diversity (species richness and biodiversity index) and stand
characters (stem number, average DBH, H, BA, crown width, crown coverage, the proportion of two
functional groups (deciduous and evergreen broadleaved species) to total stand BA). Stepwise
regression showed that BA, crown coverage, crown width, the proportion of deciduous species to total
stand BA and forest types significantly influenced LAI values measured in January, whereas species
richness, stem number, crown coverage and forest types significantly affected LAI values measured in
July (Tables S2). However, residuals of stepwise regression did not meet the requirements of normal
distribution and homogeneity (Fig. S1 and Fig. S2). The generalised additive models (GAMs) have the
advantages to analyse complex and nonlinear relationships (Guisan et al., 2002; Austin, 2002; Wood,
2006). Therefore, we used GAMs to examine how the factors selected by stepwise regression affect LAI



values. The function of GAMs is the addition of many smooth functions and each smooth function has
an explanatory variable. The variance inflation factor (VIF), the ratio of the variance of regression
coefficient for a variable when fitting with all variables to the variance of regression coefficient for the
variable if fit on its own, was used to test the multi-collinearity of explanatory variables (James et al.,
2013). When the VIF of an explanatory variable is between 0 and 10, the variable was retained to the
model; otherwise, we discarded the variable (Shen et al., 2015). The Akaike information criterion (AIC)
or generalised cross validation (GCV) was used to determine whether the model was good or bad (Clark,
2013). The factors selected after multi-collinearity test were used for multi-factor analysis. After all the
possible models in multi-factor analysis, we determined the optimal model based on the significant
influence of all explanatory variables in the model with the smallest AIC or GCV (Dong et al., 2012).
Geostatistics analysis was performed with GS+ software (Gamma Design Software). Statistical analysis
and GAMs analysis were operated in R programme. The car packages were used to test
multi-collinearity and the gam packages were used to select the optimal model.

## 3 Results

### 3.1 Variations of LAI values in three forests

The LAI values varied with forest types and measurement seasons (Table 1). Generally, LAI differed
significantly for measurement seasons ($p<0.001$), but LAI difference was not significant among forest
types ($p>0.05$). Interactive effects of measurement seasons and forest types on LAI were significant





(*p*<0.01). Among three forests, LAI in the *P. massoniana - L. glaber* forest had relatively low variations,
while LAI in the *L. glaber - C. glauca* forest had the highest variations. In the *P. massoniana - L. glaber*
forest, LAI showed the largest variations (the highest CVs) in October and the lowest variations (the
smallest CVs) in January. In the *C. axillaris* forest, the largest variation in LAI was found in April and
the lowest was found in January. In the *L. glaber - C. glauca* forest, LAI showed the largest variations
in April and had the lowest variations in July.

Mean LAI values in the three forests showed different seasonal variation patterns (Fig. 1). The *C.*

*axillaris* forest exhibited a unimodal pattern of seasonal variations, with the maximum mean LAI value
(3.11±1.18) occurring in July and the minimum mean LAI value (1.28±0.44) in January. In the *P.*
*massoniana - L. glaber* forest and *L. glaber - C. glauca* forest, the maximum mean LAI value both
occurred in October and the minimum mean LAI value appeared in January. During the growing season
(April and July), the *C. axillaris* forest had the highest mean LAI value and the *L. glaber - C. glauca*
forest had the lowest mean LAI value. During the non-growing season (October and January), the *L.*
*glaber - C. glauca* forest had the highest mean LAI value in January, while the *P. massoniana - L.*
*glaber* forest had the highest mean LAI value in October, the *C. axillaris* forest had the lowest mean
LAI values.

Mean *α* values in the three forests showed different seasonal variation patterns (Table 2). The *C.*

*axillaris* forest exhibited a unimodal pattern of seasonal variations in mean *α* value, with the maximum
mean *α* value occurring in January and the minimum mean *α* value in July. No obvious seasonal



variations were found for the mean $\alpha$ value in the *P. massoniana - L. glaber* forest and in the *L. glaber -*
*C. glauca* forest. Mean $\Omega_E$ values in the three forests were between 0.84-0.92, but they did not show
clear seasonal variations, and the standard deviations were small.

**3.2 Spatial heterogeneity of LAI values in three forests**

The semivariograms results for LAI in three forests during different measurement seasons were
summarised in Table 3. The spatially dependent variance [$C$] accounted for 88.9% - 98.4% of the total
variance [$C+C_0$] for LAI values measured in January in the three forests and also in April, July and
October in the *L. glaber - C. glauca* forest. This indicated the strong spatial autocorrelations of LAI
values at short distance. These LAI data were best fitted with gaussian model or exponential model ($r^2 >$

0.50).

Spatial autocorrelation range of LAI values differed among forests and measurement seasons (Table
3). In January, the largest spatial autocorrelation range was found in the *P. massoniana - L. glaber* forest,
and the lowest was found in the *C. axillaris* forest. In April, the largest spatial autocorrelation range of
LAI was found in the *C. axillaris* forest, and the lowest was found in the *P. massoniana - L. glaber*
forest. In July, the *P. massoniana - L. glaber* forest had the largest spatial autocorrelation range of LAI,
while the *C. axillaris* forest had the smallest spatial autocorrelation range. In October, the *L. glaber - C.*
*glauca* forest had the largest spatial autocorrelation range of LAI, while the *P. massoniana - L. glaber*
forest had the smallest spatial autocorrelation range. Seasonal changes of range showed one peak



pattern for *C. axillaris* forest and *L. glaber - C. glauca* forest, where the large range appeared in the
growing season (April and July) and the small range appeared in the non-growing season (October and
January).

Spatial distribution pattern of LAI values also varied with forest types and measurement seasons (Fig.

2). For example, LAI values in January in three forests exhibited obvious patch and heterogeneous
spatial distribution. In April and July, less spatial heterogeneity was found for LAI values in three
forests especially in the *P. massoniana - L. glaber* forest. In October, heterogeneous and patch spatial
distributions of LAI values appeared in the *L. glaber - C. glauca* forest, and banded spatial distributions
of LAI values obviously appeared in the *C. axillaris* forest.

**3.3 Factors affecting LAI variations in three forests**

The multi-collinearity test indicated that the factors with significant effects on LAI in January and

July selected by stepwise regression did not have multi-collinearity. Thus, BA, crown coverage, crown
width, the proportion of deciduous species to total stand BA and forest types were included as
explanatory variables in multi-factor analysis for LAI values measured in January in three forests (Table
4). The best fitted GAMs for LAI values in January was expressed as LAI ~ s(BA, 2) + s(crown
coverage, 2) + s(crown width, 2) + s (the proportion of deciduous species to total stand BA, 2) +factor
(forest types). For LAI values measured in July, species richness, crown coverage, stem number and
forest types were included as explanatory variables in multi-factor analysis (Table 4). The best fitted





GAMs for LAI values in July was expressed as LAI ~ s(species richness, 2) + s(crown coverage, 2) +
s(stem number, 2) + factor(forest types).
The explanatory variables included in GAMs reflected their effects on or relationship with LAI
variations. Given that other variables were fixed, LAI measured in January tended to increase as BA
increased and showed a negative nonlinear relationship with crown coverage and a positive nonlinear
relationship with crown width. The LAI values tended to decrease as the proportion of deciduous
species to total stand BA increased (Fig. 3). Given that other variables were fixed, LAI measured in July
tended to increase as species richness increased and showed a positive nonlinear relationship with
crown coverage and a negative nonlinear relationship with stem number (Fig. 4).

## 295 4 Discussion

### 296 4.1 Seasonal variations in LAI values among three forests

LAI data in subtropical forests in southern China are less than that in other global regions (Asner et
al., 2003). This study provided seasonal LAI data in three subtropical forests that consist of contrasting
functional types of species. Mean LAI values in the three forests investigated in this study varied from
1.28±0.44 to 3.28±1.26 (Table 1). This result is close to LAI range (from 1.0 in winter to 4.0 in summer)
retrieved by remote sensing techniques from subtropical area of China during the period 2000 to 2010
(Liu et al., 2012). Compared with the LAI values estimated from allometric equations (Xiang et al.,
2016) and specific leaf area (SLA) at 40 m × 40 m quadrate plots of this study (5.29-9.19), the LAI





values measured by hemipherical photography are low but significantly correlated ($r^2$=0.40 and
$p$=0.035). Previous studies (see Lopes et al., 2015) have proved the underestimation of LAI using
hemipherical photography. However, the hemipherical photography method is feasible to obtain LAI
data in forests and to investigate spatial and seasonal variations in LAI in forests (Coops et al., 2004;
Dovey & Toit, 2006).
The ratio of woody to total area ($\alpha$) and the clumping index ($\Omega_E$) have been recognised as the error
sources in LAI measurement by optical methods (Liu et al., 2015a; Bréda, 2003; Chen et al., 1997). So
far these two parameters have been measured in Northeastern China (Liu et al., 2015a; Liu et al., 2015b)
but they are not suitable for LAI correction in subtropical forests. Also the literature about them in
subtropical forests has been rarely reported. Our results showed that mean $\alpha$ values in the three forests
varied from 0.04±0.03 to 0.15±0.09 (Table 2). The variations of $\alpha$ value are probably due to the seasonal
variations and spatial heterogeneity of canopy structure in the three forests. In general, the $\alpha$ values are
consistent with the amount of leaf litter. Our results showed that the large mean $\alpha$ values occurred in
autumn for the *P. massoniana - L. glaber* forest and the *C. axillaris* forest, but in spring and autumn for
the *L. glaber - C. glauca* forest (Table 2). This seasonal change of mean $\alpha$ values in three forests was
generally consistent with the amount of leaf litter collected by litter tap installed in the three forests
(Guo et al., 2015). The average $\Omega_E$ value (0.87) in this study is smaller than the values of mixed
broadleaved - korean pine forest in Northeastern China (Liu et al. 2015b) and this could be attributed to
the different region and forests. The values of $\alpha$ and $\Omega_E$ obtained in this study fill the gap of calibration
for optical measurement of LAI in subtropical forests.
Mean LAI values differed among the three forests and the differences were significant between the *C.*
*axillaris* forest and other two forests at a given measurement season. The *C. axillaris* forest had the
relatively high mean LAI value during the growing season but changed to the lowest mean LAI value
during the non-growing season. The change in mean LAI values in the *C. axillaris* forest was consistent
with the study of a deciduous species dominated forest reported by Naithani et al. (2013). It has been
reported that the forests consisting of different plant functional types of species showed different LAI
values (Asner et al., 2003; Iio et al., 2014). The differences and seasonal variations of LAI values in the
three forests could be attributed to floristic composition and phenological defoliation pattern of tree
species especially the deciduous species. The *C. axillaris* forest consisted of 74.15% deciduous species,
25.80% evergreen broadleaved species and 0.05% evergreen coniferous species while the proportions of
deciduous species were 10.05% and 25.70% respectively in the *P. massoniana - L. glaber* forest and the
*L. glaber - C. glauca* forest. Seasonal growth and defoliation of different functional types of species
lead to the change in leaf lifespan and foliage area (Niinemets et al., 2010) during different seasons
related to temperature and water availability, which are responsible for the unimodal pattern of seasonal
variations in mean LAI values. This agrees with the results from Liu et al. (2012) that the highest LAI
was found in summer (July), followed by autumn (October) and spring (April), and the lowest was
found in winter (January).




**4.2 Spatial heterogeneity and its controlling factors of LAI values within a forest**


Semivariograms of LAI values in the three forests were fitted with spherical models, gaussian models,
exponential models or linear models (Table 3). Based on the fitted models, the degree of spatial
autocorrelation could be evaluated. Spatial autocorrelation is weak when the determination coefficient
($r^2$) of the best-fitted semivariogram model is less than 0.5 (Duffera et al., 2007). The ratio $[C/(C_0+C)]$
is also used to describe the degree of spatial autocorrelation. The ratio between 0 and 0.25 indicates a
weak spatial autocorrelation, between 0.26 to 0.75 means moderate and larger than 0.75 means strong
(Lopez-Granados et al., 2004). Spatial autocorrelation of LAI in this study varied with forests and
measurement seasons (Table 3). Strong spatial autocorrelation at short range for LAI values measured in
January in three forests indicated the sampling distance is reasonable for LAI variables within the
spatial range (Liu et al., 2008). On the contrary, weak autocorrelation within spatial autocorrelation
range indicated that more samples and smaller sampling intervals should be taken to determine spatial
dependency of LAI, such as LAI measured in April in the *P. massoniana - L. glaber* forest.
Spatial heterogeneity of LAI values was different in three forests and measurement seasons. Our
study respectively described spatial variations of LAI values by CV and analysis of geostatistics and the
results were basically consistent with each other. In general, the CVs of LAI values in three forests (in
particular *C. axillaris* forest) were higher for the period of leaf onset (April) and senescence (October)
than that for the period of leaf maturity (July) (Table 1). This reflects changes of leaves due to plant
phenology and is consistent with the study by Naithani (2013) that LAI became increasingly



homogenous from leaf onset to maturity, but became more heterogeneous from maturity to senescence.
As a result, heterogeneity degree of LAI values in three forests tended to dwindle from leaf
non-growing season to growing season (Fig. 2).
The complex hydrothermal environment results in complex vertical and horizontal variations of
canopy layer and formed the unique spatial heterogeneity of LAI values. The results of stepwise
regression and GAMs showed that forest types, species diversity and stand characters affected the
spatial heterogeneity of LAI values significantly in three forests. This finding that floristic composition
and stand characters affected LAI values measured in July is consistent with the previous study that LAI
values increased with species richness (Yao et al., 2015). The positive relationship between LAI values
in July and crown coverage is also in agreement with the findings reported by Bequet (2012) that large
trees (high DBH, tree height, crown length and crown cover) had high LAI values measured in July. The
negative relationship between LAI values in July and stem number probably could be explained by the
strong competition among tree species in the three forests with diverse species composition.
Up to now, the relationship of LAI variations measured in non-growing season with forest types and
stand characters has seldom reported. In this study, forest types, BA, crown width, crown coverage and
the proportion of deciduous species to total stand BA were the factors significantly affecting LAI
variations in January. Because large trees (high crown width and BA) had high LAI values and January
is the leaf senescence period of deciduous species, LAI values in January increased with BA and crown
width but decreased with deciduous species ratio. The fact that LAI values in January decreased with



increasing crown coverage could be explained that large crown coverage means more defoliation (in
particular deciduous species) in the forest in January.
Although the factors selected by regression could explain a small proportion (6%) of spatial
heterogeneity of LAI measured in July, the factors selected in January could explain 35% of the LAI
spatial heterogeneity (Table 4). The LAI heterogeneity also could be affected by that several other
factors, such as the topography (Naithani et al., 2012), soil features (Chloer et al., 2010), soil
temperature (Vitasse et al., 2009; Hardwick et al., 2015), microclimate, human activities and other
physicochemical properties. And yet, full leaf expansion of all tree species which covers up the effect of
other physicochemical properties on LAI leads to small difference in LAI in July. The effects of
environmental factors (such as temperate, rainfall, etc.) on LAI in the forests at fine scale should be
taken into account in the future studies.
Spatial heterogeneity of LAI in three forests can yield some useful information for sampling strategy
to accurate estimation of LAI by using indirect measurement. An optimal sampling strategy should
consider appropriate sampling plot size and the lowest sampling number as far as possible to obtain a
high sampling accuracy and a low sampling error (Bequet et al., 2012). Our study found that strong
spatial autocorrelations range were about 30m (Table 3), indicating that 30m range might serve as a
reference for sampling plot size to estimate LAI in subtropical forests. In addition, LAI heterogeneity
was closely related to floristic composition and stand characters, thus stand structural variables (BA or
DBH) are important for sampling strategy to measure LAI in forests (Bequet et al., 2012).

## 5 Conclusions

This study measured LAI in three subtropical forests using hemispherical photography method in
four seasons and offered reliable data to analyse spatial and seasonal variations of LAI in the three
forests. Our results indicated that LAI differed greatly with forests and measurement seasons. Seasonal
variations of LAI occurred in the three forests reflect defoliation phenomenon due to plant phenology.
LAI values in the three forests exhibited different spatial autocorrelation in four seasons. Obvious patch
distribution pattern of LAI values was found in the three forests during the non-growing seasons and
this pattern gradually dwindled in the growing seasons. While BA, crown coverage, crown width, the
proportion of deciduous species to total stand BA and forest types significantly affected the spatial
variations in LAI values in January, species richness, crown coverage, stem number and forest types
significantly affected the spatial variations in LAI values in July. These findings supplement LAI data
for global synthesis and provide useful information for sampling strategies to accurate LAI estimates
and simulating the models of forest production and hydrological cycle in subtropical forests.

## Acknowledgements

This study was supported by the Specialized Research Fund for the Doctoral Program of Higher
Education (20124321110006), the National Natural Science Foundation of China (31570447 and
31300524), the Programme of State Forestry Special Fund for Public Welfare Sectors of China



(201304317), and the New Century Excellent Talents Program (NCET-06-0715). Thanks also go to
the staff of the administration office of Dashanchong Forest Farm, Changsha County, Hunan Province,
for their local support.

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

with English abstract)














**Table 1** Descriptive statistical characteristics of LAI values measured during period from April 2014 to
January 2015 in *P. massoniana - L. glaber*, *C. axillaris* and *L. glaber - C. glauca* forests (n=100).

| Month | Forest type | Minimum value | Maximum value | Variance coefficient (%) | *P* value of K-S test | Data transformation |
|---|---|---|---|---|---|---|
| January | *P. massoniana - L. glaber* | 1.29 | 4.03 | 27.5 | 0.021 | 0.275 |
| | *C. axillaris* | 0.53 | 2.38 | 34.0 | 0.260 | |
| | *L. glaber - C. glauca* | 0.43 | 6.98 | 40.2 | 0.018 | 0.243 |
| April | *P. massoniana - L. glaber* | 1.57 | 7.83 | 36.4 | 0.076 | |
| | *C. axillaris* | 1.34 | 8.33 | 47.0 | 0.047 | 0.535 |
| | *L. glaber - C. glauca* | 1.34 | 10.22 | 59.6 | 0.000 | 0.158 |
| July | *P. massoniana - L. glaber* | 1.56 | 8.16 | 38.0 | 0.003 | 0.075 |
| | *C. axillaris* | 1.73 | 8.17 | 37.8 | 0.166 | |
| | *L. glaber - C. glauca* | 1.68 | 7.58 | 33.1 | 0.010 | 0.170 |
| October | *P. massoniana - L. glaber* | 1.55 | 6.79 | 38.3 | 0.321 | |
| | *C. axillaris* | 0.37 | 6.51 | 44.1 | 0.102 | |
| | *L. glaber - C. glauca* | 1.49 | 7.88 | 49.3 | 0.000 | 0.212 |






**Table 2** Average woody to total leaf ration ($\alpha$) and clumping index ($\Omega_E$) values in *P. massoniana - L. glaber*, *C. axillaris* and *L. glaber - C. glauca* forests. Values in parenthesis are the standard deviation of $\alpha$ and $\Omega_E$ values (n=100).

| Forest type | Month | $\alpha$ | $\Omega_E$ |
|---|---|---|---|
| *P. massoniana - L. glaber* | January | 0.06 (0.04) | 0.88 (0.09) |
| | April | 0.08 (0.05) | 0.87 (0.09) |
| | July | 0.07 (0.04) | 0.87 (0.09) |
| | October | 0.09 (0.10) | 0.85 (0.08) |
| *C. axillaris* | January | 0.15 (0.09) | 0.92 (0.08) |
| | April | 0.07 (0.06) | 0.85 (0.10) |
| | July | 0.04 (0.03) | 0.90 (0.07) |
| | October | 0.14 (0.14) | 0.87 (0.10) |
| *L. glaber - C. glauca* | January | 0.07 (0.09) | 0.87 (0.09) |
| | April | 0.15 (0.07) | 0.86 (0.09) |
| | July | 0.05 (0.03) | 0.87 (0.08) |
| | October | 0.09 (0.08) | 0.84 (0.09) |



**Table 3** Semivariogram theoretical models and fitted parameters for LAI values in *P. massoniana - L.*
*glaber*, *C. axillaris* and *L. glaber - C. glauca* forests.

| Month | Forest type | Model | Nugget $(C_0)$ | Sill $(C_0+C)$ | $C/(C_0+C)$ | Range $(A_0/m)$ | $r^2$ | RSS |
|---|---|---|---|---|---|---|---|---|
| January | *P. massoniana - L. glaber* | Exponential | 0.0068 | 0.0614 | 0.889 | 27.00 | 0.607 | $9.762 \times 10^{-5}$ |
| | *C. axillaris* | Exponential | 0.0030 | 0.1820 | 0.984 | 13.80 | 0.504 | $1.219 \times 10^{-4}$ |
| | *L. glaber - C. glauca* | Gaussian | 0.0029 | 0.1178 | 0.975 | 15.42 | 0.888 | $3.468 \times 10^{-5}$ |
| April | *P. massoniana - L. glaber* | Exponential | 0.1220 | 0.7670 | 0.841 | 17.70 | 0.229 | 0.017 |
| | *C. axillaris* | Linear | 0.1760 | 0.1760 | 0.000 | 52.96 | 0.189 | $1.762 \times 10^{-4}$ |
| | *L. glaber - C. glauca* | Exponential | 0.0008 | 0.0152 | 0.951 | 26.40 | 0.978 | $2.290 \times 10^{-7}$ |
| July | *P. massoniana - L. glaber* | Linear | 0.0843 | 0.0843 | 0.000 | 92.69 | 0.074 | $1.383 \times 10^{-4}$ |
| | *C. axillaris* | Exponential | 0.1460 | 0.9340 | 0.844 | 17.70 | 0.258 | 0.017 |
| | *L. glaber - C. glauca* | Exponential | 0.0065 | 0.0684 | 0.905 | 22.80 | 0.951 | $5.781 \times 10^{-6}$ |
| October | *P. massoniana - L. glaber* | Exponential | 0.1620 | 1.6310 | 0.901 | 11.70 | 0.173 | 0.017 |
| | *C. axillaris* | Spherical | 0.0050 | 0.5830 | 0.991 | 11.90 | 0.000 | $1.870 \times 10^{-3}$ |
| | *L. glaber - C. glauca* | Exponential | 0.0005 | 0.0125 | 0.960 | 21.90 | 0.894 | $4.444 \times 10^{-7}$ |









**Table 4** Estimated coefficients of the generalised additive models (GAMs) for the factors with effects

on LAI values measured in *P. massoniana - L. glaber*, *C. axillaris* and *L. glaber - C. glauca* forests.

| Month | Parameter | *F*-value | *p*-value | $r^2$ | AIC |
|---|---|---|---|---|---|
| January | s (BA,2) | 15.146 | $1.236 \times 10^{-4}$*** | 0.3496 | 654.30 |
| | s (Crown width,2) | 1.588 | 0.209 | | |
| | s (Crown coverage,2) | 0.556 | 0.456 | | |
| | s (PDSB,2) | 49.324 | $1.556 \times 10^{-11}$*** | | |
| | factor(Forest types) | 44.355 | $< 2.2 \times 10^{-16}$*** | | |
| July | s (Species richness,2) | 3.165 | 0.076. | 0.0551 | 677.85 |
| | s (Stem number,2) | 8.381 | $4.086 \times 10^{-3}$** | | |
| | s (Crown coverage,2) | 0.006 | 0.939 | | |
| | factor(Forest types) | 2.475 | 0.086. | | |

The significance of the regressions (*p*) are ., *, **, *** for *p*<0.1, 0.05, 0.01, and 0.001, respectively



## Figure captions

**Fig. 1** Seasonal variations of mean LAI values (with standard deviation) in *P. massoniana - L. glaber*, *C. axillaris* and *L. glaber - C. glauca* forests. The values carrying with different letters indicate significant differences ($p<0.05$) among measurement seasons in a given forest.

**Fig. 2** Spatial heterogeneity map of LAI values interpolated through ordinary Kriging method for *P. massoniana - L. glaber*, *C. axillaris* and *L. glaber - C. glauca* forests.

**Fig. 3** Partial effects of total stand basal area (BA, cm$^2$), average crown width (m), crown coverage (m$^2$), the proportion of deciduous species to total stand BA (PDSB), and forest types (calculated for overstory trees with height larger than average stand height) on the LAI values observed in January in *P. massoniana - L. glaber*, *C. axillaris* and *L. glaber - C. glauca* forests.

**Fig. 4** Partial effects of species richness, individual stem number, crown coverage (m$^2$) and forest types (calculated for overstory trees with height larger than average stand height) on the LAI values observed in July in *P. massoniana - L. glaber*, *C. axillaris* and *L. glaber - C. glauca* forests.

**Figure 1**

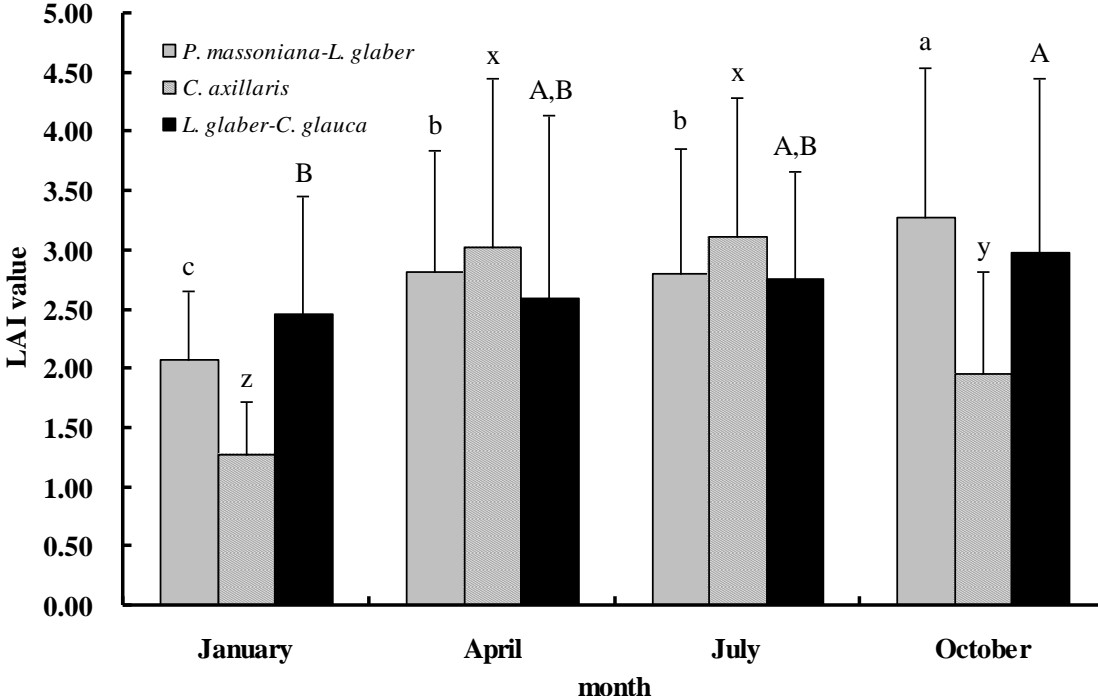












**Figure 2**






**Figure 3**




**Figure 4**

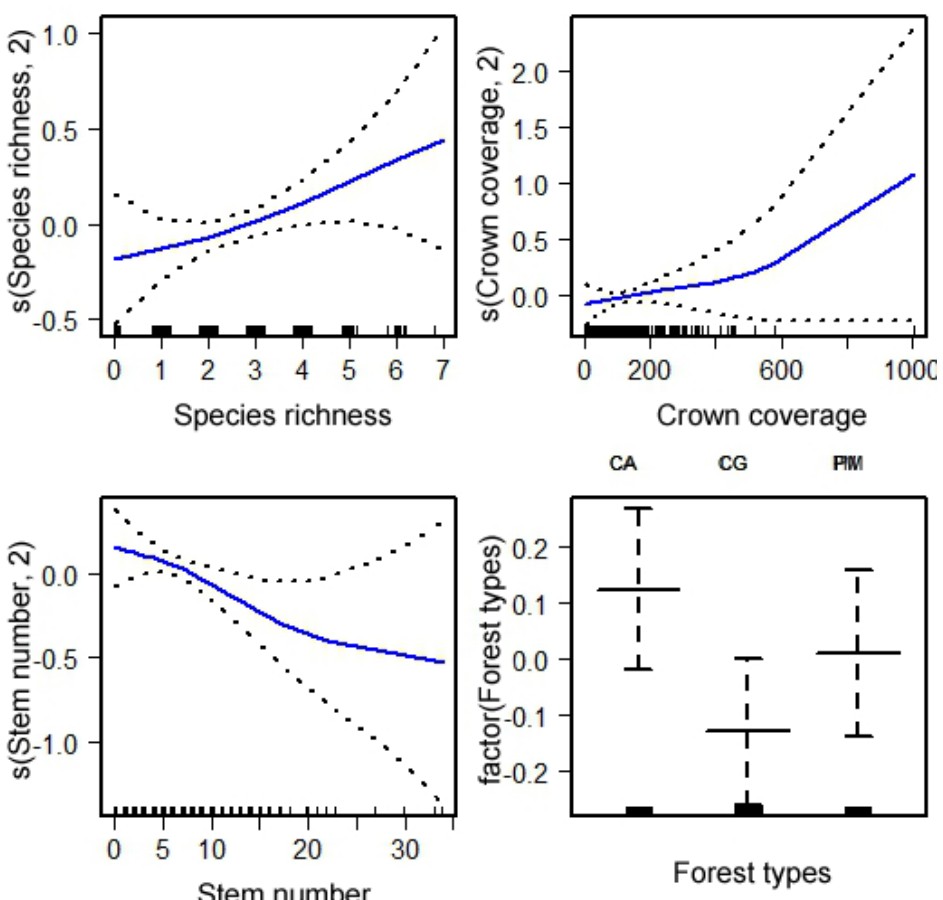



