# Peer review of "Spatial and seasonal variations of leaf area index (LAI) in subtropical secondary"

_Biogeosciences, 2016_

## Referee Comment (RC1) · Anonymous Referee #1 · 27 Feb 2016

The authors investigated seasonal variation, spatial heterogeneity of LAI and its controlling factors by using spatial statistics and generalized additive models (GAM) based on observed values of three forests in subtropical China. They found that LAI values differed greatly by forest types and seasons and showed strong spatial autocorrelation. Species diversity and stand variables like stand density affected LAI values. The work is new for subtropical forests. This is a well-written manuscript well suited for biogeoscience. The topic is of general interest to readers in the field of forest ecosystem process. I only have a few questions/comments on model part.

1. The authors mentioned they used GAM followed by linear step regression (LSR). You may directly use GAM for stepwise regression by MGCV packages in R and not

necessary perform LSR since GAM could describe both linear and nonlinear relationship. 2. In the methods, you need to report which smooth method you used for GAM. 3. For the results of model fitting, you listed some variables which were not statistically significant (p>0.1). For example, BA(p=0.258), crown width(p=0.327) and crown coverage (p=0.333) in Table S1 for LSR and crown width (p=0.209) and crown coverage (p=0.456) for GAM in Table 2. This will change the conclusion on the variables related with LAI. Although the model is not for prediction, you may lower the significant level. Please carefully check the results. 4. Page 11 Line 11. "Tree species diversity" is better than "species diversity". 5. Page 21 Lines 395-396. It is interesting the authors recommended 30m as a reference for sampling plot size to estimate LAI in subtropical forests. However, you may use a range not point value to account this according to table 3. 6. As the author mentioned, there are many factors affecting LAI. As an important stand structure characteristic, stand structural diversity (tree size diversity in this case) may explain LAI variation partially. I suggest testing the factor in the study. 7. Fig. 1. P. massoniana-L. glaber and C. axillaris cannot be recognized clearly. Please change the legend. 8. Fig.3. I am wondering you may have wrong values for BA (range from 0 to 6000?) and crown coverage (range from 0 to 1000?). What is the unit for them? Same as Fig. 4. Please carefully check them. 9. Table S1. The summary of values of stem density, BA and IV by species are not equal to the whole stand. 10. Table S2. Parts of the columns of mean sq and sum aq are the same? Actually you need not to report these values besides parameters, F values and p values.

---

## Short Comment (SC1) · 6 Mar 2016

The manuscript entitled "Spatial and seasonal variations of leaf area index (LAI) in subtropical secondary forests related to floristic composition and stand characters" by Zhu et al. is an interesting study on the spatial heterogeneity of LAI and its controlling factors in subtropical forests in China. The paper covers an important issue. The investigation is in-depth and thorough. The results are interesting and fill the gap of LAI measurement in subtropical forests. The paper is well-written and duly illustrated. Publication is therefore recommended with minor revisions suggested as follows:

1. Line 33-34: insert a word "and" after the geostatistics method. 2. Line 46: remove the keywords "Deciduous species". In your paper, more than one tree species were

investigated and the constituents of forests or tree species richness was one of the controlling factors of LAI values. In other words, "deciduous species" is not a proper substitute for the proportion of deciduous species. 3. Line 51, 55-56 and throughout main text, the reference should be arranged by the published year. 4. Line 59: insert a word "as" between "used" and "parameter". 5. Line 109: change "stand character" to "stand characters". 6. Line 133-134: the mean temperature of the study site should be a fixed value, please correct it. 7. Line 148: check and correct the plot size of P. massoniana - L. glaber mixed forests. 8. Line 170: please add the manufacturer and country to the LAI measuring instrument (SY-S01A). 9. Line 199-200: coefficient of variation (CV) does not need full name here. 10. Line 234-238: the author need to report which smooth method used for GAM in this study. 11. Line 250: it is better to illustrate the version of R software used in this study. 12. Line 255: consider changing "month" in Table 1 into "measurement seasons". Do the same modifications in other tables and Fig. 1. 13. Line 265: How did you calculate the mean LAI values? I'm a little confused that why you think it's necessary to report the minimum, maximum and mean values of LAI at the same time. what's the differences or the particular meaning between them? 14. Line 350: ".... but they are not suitable for LAI correction in subtropical forests", why? Is this a conclusion drew by yourself or from other's research? 15. Line 360: change "is" to "was". 16. Line 695-700: the "RSS" in the first line in Table 3 need to be clarified. 17. Line 745-750: In Fig.1, the y-axis should change into "mean LAI value", x-axis should change into "Month". 18. Fig 3 and 4: These two figures are new and unique, and the results might be interesting. It's a pity that you didn't thoroughly discuss these figures except simply described in Results Line 326-330. I suggest to add some discussion about these two figures in you manuscript.

---

## Referee Comment (RC2) · Anonymous Referee #2 · 15 Apr 2016

**GENERAL COMMENT**

The topic of the manuscript lies within the field of the journal Biogeosciences. It reports on spatial and temporal variability of the leaf area index in forests. The overall importance of reliable LAI measurements is undoubted and systematic studies of spatial variability within forest stands are seldom. In this sense, the present study is justified. Unfortunately, the description of the methods is insufficient and the obtained results remain therefore questionable.

**DETAILED COMMENTS**

Material and methods

Since this determines the canopy structure, it should be stated if the studied forest plots where planted or if they are from natural regeneration, further if they were thinned or selectively cut at some point in time. According to the supplementary table S1, it appears that the stands are uneven aged, but a clear information about their history would be useful. The material used for the hemispherical photography is only poorly described. The camera type is given, but not its manufacturer. There is no information about the lens, not even its viewing angle (or focal length). The choice of picture exposure is not described although it is essential to achieve a good contrast without overexposure. The resolution of the pictures is not given, nor their format. The picture analysis is also insufficiently described. There is no indication of the software used, of the pixel classification (thresholding), of the considered viewing angle and if it was divided into rings. The viewing angle would be very important to know here because, in conjunction with the tree height, it determines the integration area of the LAI measurement (which is, in turn, important for understanding the spatial variability). The method to estimate a clumping factor does not state the number of sectors used. The estimation of the contribution of leaves versus wood to the plant area index would be a positive aspect of this study, but here also the methods are poorly described. It is not stated if all of the woody elements on all pictures were painted or only sub-samples. Further, "replace the woody materials with surrounding of non-woody materials" is either a wrong wording or a wrong method. Woody areas should neither be replaced by "non-woody materials" nor by sky pixels, they should be excluded from the analysis because it is essentially not known how much leaf area or sky area they hide. Statistical tests are partly done after different types of data transformation. I'm not sure if cutting outliers back to "normally maximal values" is an appropriate method, but at least the measure of this transformation in table 1 should be described in an understandable manner. Using non-parametric statistics would probably make the tests more convincing than the different transformations applied here. Crown coverage is used as a factor in statistical models, but it is not described what this parameter means and how it was measured. A crown coverage is often derived from hemispherical photographs. Is it also the case

here, or is it an independent measurement? This can completely change the interpretation of the obtained statistical relationship. The kriging is also insufficiently described in the methods section (it is only in a figure legend that it is given as "ordinary"). The maps produced by this kriging show island structures that probably correspond to the grid of picture taking. If this is true, then it indicates a methodological problem. Either the photographs were systematically taken in some spatial relation to the trees (e.g. on a regular grid in a regularly planted stand). Or the very goal of kriging, i.e. interpolating between discrete measurements, was missed.

Results and discussion

The presented results would probably be interesting, but due to the poor description of the methods they are all more of less doubtful.

Tables and figures

Table 1 and 2 should use the same structure to be comparable. Table 3 should include the sample size, otherwise the column RSS is meaningless Table 4 gives statistical tests without giving any information on how the different factors affect the dependent variable. Since this is not so easy to put in a table in the case of non-linear relationships, table 4 should make a reference to fig. 3. Figure 1: the two grey tones cannot be distinguished.

Language

The English of the manuscript is well understandable but some sentences are not well structured. At least in one case the wording is inappropriate: "throughout four measurement seasons" would mean at least several measurements in each season (while there is actually one per season).
* * *

---

## Author Comment (AC1) · 27 Apr 2016

The manuscript entitled "Spatial and seasonal variations of leaf area index (LAI) in subtropical secondary forests related to floristic composition and stand characters" by Zhu et al. is an interesting study on the spatial heterogeneity of LAI and its controlling factors in subtropical forests in China. The paper covers an important issue. The investigation is in-depth and thorough. The results are interesting and fill the gap of LAI measurement in subtropical forests. The paper is well-written and duly illustrated. Publication is therefore recommended with minor revisions suggested as follows: Re: First of all, we thank X. Zhang very much for the positive comments and valuable suggestions. Based on the following comments, we will revise the manuscript and our

detailed replies are presented below.

1. Line 33-34: insert a word "and" after the geostatistics method. Re: We will add "and" as suggested.

2. Line 46: remove the keywords "Deciduous species". In your paper, more than one tree species were investigated and the constituents of forests or tree species richness was one of the controlling factors of LAI values. In other words, "deciduous species" is not a proper substitute for the proportion of deciduous species. Re: "Deciduous species" will be deleted from the keywords.

3. Line 51, 55-56 and throughout main text, the reference should be arranged by the published year. Re: Based on the comments, all references in the entire manuscript will be re-arranged according to a chronological order.

4. Line 59: insert a word "as" between "used" and "parameter". Re: Instead of adding "as", we will change the sentence into "Leaf area index (LAI), defined as total one-sided leaf area per unit ground surface area (Biudes et al., 2014), is a widely used parameter to….".

5. Line 109: change "stand character" to "stand characters". Re: We will change "stand character" to "stand characters" as suggested.

6. Line 133-134: the mean temperature of the study site should be a fixed value, please correct it. Re: We will change the mean annual air temperature into "16.5°C".

7. Line 148: check and correct the plot size of P. massoniana - L. glaber mixed forests. Re: We have checked and the plot size is correct.

8. Line 170: please add the manufacturer and country to the LAI measuring instrument (SY-S01A). Re: The LAI measuring instrument was made by Shiya Scientific and Technical Cooperation in China. We will add the manufacturer and country in the manuscript.

9. Line 199-200: coefficient of variation (CV) does not need full name here. Re: We will use abbreviation CV only.

10. Line 234-238: the author need to report which smooth method used for GAM in this study. Re: Good point. We will indicate that the smooth method for GAM is smooth spline.

11. Line 250: it is better to illustrate the version of R software used in this study. Re: We will add the version of R (R 3.2.1) in the manuscript.

12. Line 255: consider changing "month" in Table 1 into "measurement seasons". Do the same modifications in other tables and Fig. 1. Re: Change as suggested.

13. Line 265: How did you calculate the mean LAI values? I'm a little confused that why you think it's necessary to report the minimum, maximum and mean values of LAI at the same time. what's the differences or the particular meaning between them? Re: We calculated average LAI values of 100 plots in each forest at a given measurement season. The minimum and maximum values within a forest at different measurement seasons to examine the variations in LAI.

14. Line 350: "…. but they are not suitable for LAI correction in subtropical forests", why? Is this a conclusion drew by yourself or from other's research? Re: The previous studies by Liu et al. (2015a) and Liu et al. (2015b) showed that the $\alpha$ values ranged from $0.04 \pm 0.01$ to $0.69 \pm 0.12$ and $\Omega E$ values ranged from $0.88 \pm 0.04$ to $0.96 \pm 0.01$. These values were measured in temperate forest in northeastern China and differed from our study ($\alpha$ ranged from $0.04 \pm 0.03$ to $0.15 \pm 0.09$ and $\Omega$ ranged from $0.84 \pm 0.09$ to $0.92 \pm 0.08$). Therefore, we drew the conclusion and revised the sentence.

15. Line 360: change "is" to "was". Re: Change as suggested.

16. Line 695-700: the "RSS" in the first line in Table 3 need to be clarified. Re: We will offer the full name of RSS (residual sum of squares).

17. Line 745-750: In Fig.1, the y-axis should change into "mean LAI value", x-axis

should change into "Month". Re: Changed as suggested.

18. Fig 3 and 4: These two figures are new and unique, and the results might be interesting. It's a pity that you didn't thoroughly discuss these figures except simply described in Results Line 326-330. I suggest to add some discussion about these two figures in you manuscript. Re: Good comments. We will add some sentences to discuss the results of Fig. 3 and Fig. 4.

---

## Author Comment (AC2) · 12 May 2016

We thank the anonymous referee for the valuable comments and suggestions. We will carefully take the suggestions into consideration when we revise the manuscript. Our detailed responses to the comments are presented below.

GENERAL COMMENT The topic of the manuscript lies within the field of the journal Biogeosciences. It reports on spatial and temporal variability of the leaf area index in forests. The overall importance of reliable LAI measurements is undoubted and systematic studies of spatial variability within forest stands are seldom. In this sense, the present study is justified. Unfortunately, the description of the methods is insufficient and the obtained results remain therefore questionable. Re: Thanks for the positive

[Figure]

comments and suggestions on our manuscript. Based on the comments, we will revise the methods and results sections. The detailed responses are presented as follows.

DETAILED COMMENTS Material and methods Since this determines the canopy structure, it should be stated if the studied forest plots where planted or if they are from natural regeneration, further if they were thinned or selectively cut at some point in time. According to the supplementary table S1, it appears that the stands are uneven aged, but a clear information about their history would be useful. Re: The forests are from natural regeneration and no thinning or selectively cutting were applied there till to investigation. We will describe the history of the forests in the revision.

The material used for the hemispherical photography is only poorly described. The camera type is given, but not its manufacturer. There is no information about the lens, not even its viewing angle (or focal length). The choice of picture exposure is not described although it is essential to achieve a good contrast without overexposure. The resolution of the pictures is not given, nor their format. Re: Thanks for valuable suggestion. We will provide clear information about the material of the hemispherical photography, such as the manufacturer (Shiya Scientific and Technical Cooperation, China), the lens (Pentax TS2V114E, Japan), the viewing angle (180°), the picture exposure (automatic exposure set by the manufacturer), the picture resolution and format (768×494 pix, BMP).

The picture analysis is also insufficiently described. There is no indication of the software used, of the pixel classification (thresholding), of the considered viewing angle and if it was divided into rings. The viewing angle would be very important to know here because, in conjunction with the tree height, it determines the integration area of the LAI measurement (which is, in turn, important for understanding the spatial variability). Re: We will revise the manuscript by adding the description of the picture analysis such as the software (the plant canopy analysis software developed by the manufacturer), the pixel classification (thresholding) (752(H) ×582(V)), the considered viewing angle (150°) and it was divided into 5 rings.

The method to estimate a clumping factor does not state the number of sectors used. The estimation of the contribution of leaves versus wood to the plant area index would be a positive aspect of this study, but here also the methods are poorly described. It is not stated if all of the woody elements on all pictures were painted or only sub-samples. Further, "replace the woody materials with surrounding of non-woody materials" is either a wrong wording or a wrong method. Woody areas should neither be replaced by "non-woody materials" nor by sky pixels, they should be excluded from the analysis because it is essentially not known how much leaf area or sky area they hide. Re: Sorry for our unclear description. We originally described the method according to the reference (Liu ZL, Jin GZ, Chen JM, Qi YJ. 2015. Evaluating optical measurements of leaf area index against litter collection in a mixed broadleaved-Korean pine forest in China. Trees, 29: 59-73), where the word "replace" used means "exclude the pixels of woody materials". We will change the sentence into "In Photoshop software, we used the Clone Stamp Tool to select the image of the woody materials (e.g., stems) and excluded the pixels, leaving only leaves on the photos".

Statistical tests are partly done after different types of data transformation. I'm not sure if cutting outliers back to "normally maximal values" is an appropriate method, but at least the measure of this transformation in table 1 should be described in an understandable manner. Using non-parametric statistics would probably make the tests more convincing than the different transformations applied here. Re: Yes, you are right. Our description is not clear, so we will change the sentences into "According to Chiang et al. (2003), we regarded the LAI values as the normal values when the LAI values were within mean value $\pm$ 3 $\times$ standard deviation. Otherwise, the LAI values were outliers and replaced with the maximum or the minimum of normal values. Because the geostatistics analysis requires that the data meet normal distribution, the transformation was applied if the data did not meet normal distribution". To support our method, we will cite the references (Chiang LH, Pell RJ, Seasholtz MB. 2003. Exploring process data with the use of robust outlier detection algorithms. Journal of Process Control, 13(5): 437-449; Dai FQ, Zhou QG, Lv ZQ, Wang XM, Liu GC. 2014. Spatial prediction

of soil organic matter content integrating artificial neural network and ordinary kriging in Tibetan Plateau. Ecological Indicators, 45: 184-194) in the text and add them in the reference list. We hope this revision is clearer than it was before.

Crown coverage is used as a factor in statistical models, but it is not described what this parameter means and how it was measured. A crown coverage is often derived from hemispherical photographs. Is it also the case here, or is it an independent measurement? This can completely change the interpretation of the obtained statistical relationship. Re: Sorry for our ambiguous description. The crown coverage was not derived from hemispherical photographs and it was calculated from crown diameter measured for individual trees within a stand.

The kriging is also insufficiently described in the methods section (it is only in a figure legend that it is given as "ordinary"). The maps produced by this kriging show island structures that probably correspond to the grid of picture taking. If this is true, then it indicates a methodological problem. Either the photographs were systematically taken in some spatial relation to the trees (e.g. on a regular grid in a regularly planted stand). Or the very goal of kriging, i.e. interpolating between discrete measurements, was missed. Re: According to this comments, we will add description of the Kriging in the methods section. Although the ordinary Kriging has the drawback, it is a commonly used interpolating method in the geostatistics reported by other studies (Elbasiouny H, Abowaly M, Abuïij£Alkheir A, Gad A. 2014. Spatial variation of soil carbon and nitrogen pools by using ordinary Kriging method in an area of north Nile Delta, Egypt. Catena, 113: 70-78. Dai FQ, Zhou QG, Lv ZQ, Wang XM, Liu GC. 2014. Spatial prediction of soil organic matter content integrating artificial neural network and ordinary kriging in Tibetan Plateau. Ecological Indicators, 45: 184-194).

Results and discussion The presented results would probably be interesting, but due to the poor description of the methods they are all more of less doubtful. Re: You are right. We will revise the Methods section (see our responses mentioned above) based on this comment.

Tables and figures Table 1 and 2 should use the same structure to be comparable. Table 3 should include the sample size, otherwise the column RSS is meaningless. Table 4 gives statistical tests without giving any information on how the different factors affect the dependent variable. Since this is not so easy to put in a table in the case of non-linear relationships, table 4 should make a reference to fig. 3. Figure 1: the two grey tones cannot be distinguished. Re: We will use the same structure for Table 1 and Table 2, and add the sample size in Table 3 as suggested. Based on the comments, in Table 4 we will make the reference to Fig. 3. The two grey bars in Figure 1 will be changed into empty and grey, respectively.

Language The English of the manuscript is well understandable but some sentences are not well structured. At least in one case the wording is inappropriate: "throughout four measurement seasons" would mean at least several measurements in each season (while there is actually one per season). Re: We will revise the manuscript and ask a native English editor to help improve the language.

---

## Author Comment (AC3) · 12 May 2016

We are grateful to the anonymous referee for the constructive comments and suggestions. We will carefully take the suggestions into consideration when we revise the manuscript. Our detailed responses to the comments are presented below.

The authors investigated seasonal variation, spatial heterogeneity of LAI and its controlling factors by using spatial statistics and generalized additive models (GAM) based on observed values of three forests in subtropical China. They found that LAI values differed greatly by forest types and seasons and showed strong spatial autocorrelation. Species diversity and stand variables like stand density affected LAI values. The work is new for subtropical forests. This is a well-written manuscript well suited for biogeo-

[Figure]

**BGD**

science. The topic is of general interest to readers in the field of forest ecosystem process. I only have a few questions/comments on model parts. Re: Thanks for the overall positive and valuable comments on our manuscript. Based on comments, we will revise the manuscript. Please see the detailed responses below.

1. The authors mentioned they used GAM followed by linear step regression (LSR). You may directly use GAM for stepwise regression by MGCV packages in R and not necessary perform LSR since GAM could describe both linear and nonlinear relationship. Re: It is a good point. Based on this comment, we used GAM model directly instead of fitting the model by two steps (that is LSR and GAM). Because there are two packages ("gam" and "mgcv") developed in R project for GAM and the two packages have the same function, we still used gam package for GAM analysis. Our results (see the following table and figures) showed that the factors affecting LAI variations differed slightly from the results in the previous manuscript, but the effects were significant. We will revise the manuscript accordingly. We hope our results are satisfactory for publication.

2. In the methods, you need to report which smooth method you used for GAM. Re: Yes, we will report that the smooth method for GAM is smooth spline method with two splines.

3. For the results of model fitting, you listed some variables which were not statistically significant (p>0.1). For example, BA (p=0.258), crown width (p=0.327) and crown coverage (p=0.333) in Table S1 for LSR and crown width (p=0.209) and crown coverage (p=0.456) for GAM in Table 2. This will change the conclusion on the variables related with LAI. Although the model is not for prediction, you may lower the significant level. Please carefully check the results. Re: Based on the comments above, we re-run the gam package directly and the variables with statistical significance are presented in the table (see the above table). Thus, the variables which are not significant are not shown and the results and conclusion will be revised accordingly.

4. Page 11 Line 11. "Tree species diversity" is better than "species diversity". Re: We will change "species diversity" into "tree species diversity" as suggested.

5. Page 21 Lines 395-396. It is interesting the authors recommended 30m as a reference for sampling plot size to estimate LAI in subtropical forests. However, you may use a range not point value to account this according to table 3. Re: We will replace the point value with the range value (i.e. from 13m to 27m) based on this comment.

6. As the author mentioned, there are many factors affecting LAI. As an important stand structure characteristic, stand structural diversity (tree size diversity in this case) may explain LAI variation partially. I suggest testing the factor in the study. Re: Good suggestion! We calculated the tree size diversity based on the reference (Lei XD, Wang WF, Peng CH. 2008. Relationships between stand growth and structural diversity in spruce-dominated forests in New Brunswick, Canada. Canadian Journal of Forest Research, 39, 1835-1847). Then we have added this variable to run GAM model and found no significance for it.

7. Fig. 1. P. massoniana-L. glaber and C. axillaris cannot be recognized clearly. Please change the legend. Re: We will change as suggested.

8. Fig.3. I am wondering you may have wrong values for BA (range from 0 to 6000?) and crown coverage (range from 0 to 1000?). What is the unit for them? Same as Fig. 4. Please carefully check them. Re: In the previous manuscript the unit for BA was cm2 and for crown coverage was m2. We only used the data of individual trees with height larger than average height in each stand, so some values of BA and crown coverage were within the range. After re-running the GAM model, only total crown coverage of the stand is a significant variable. We checked the data carefully and will present the right results.

9. Table S1. The summary of values of stem density, BA and IV by species are not equal to the whole stand. Re: In the previous manuscript, the data in Table S1 were for the all species and the top five tree species. The data of other species were not

provided in Table S1. Sorry for our carelessness. We will add one row to show the summed data for the rest species.

10. Table S2. Parts of the columns of mean sq and sum aq are the same? Actually you need not to report these values besides parameters, F values and p values. Re: Yes, you are right. We will delete these two columns as suggested.

**Table 4** Estimated coefficients of the generalised additive models (GAMs) for the factors with effects on LAI values measured in *P. massoniana - L. glaber, C. axillaris* and *L. glaber - C. glauca* forests.

| Measurement seasons | Parameter | *F*-value | *p*-value | $r^2$ | AIC |
|---|---|---|---|---|---|
| January | s (Stem number, 2) | 16.716 | <0.0001*** | 0.3481 | 655.91 |
| | s(Crown coverage, 2) | 4.545 | 0.034* | | |
| | s (PESB, 2) | 26.105 | <0.0001*** | | |
| | s (PDSB, 2) | 27.281 | <0.0001*** | | |
| | factor(Forest types) | 39.847 | <0.0001*** | | |
| July | s (Stem number, 2) | 5.027 | 0.026* | 0.040 | 880.93 |
| | s (PDSB, 2) | 7.115 | 0.008** | | |

**Fig. 1.**

**Fig. 3** Partial effects of stem number, crown coverage (m²), the proportion of evergreen conifer species to total stand BA (PESB), the proportion of deciduous species to total stand BA (PDSB) and forest types (calculated for overstory trees with height larger than average stand height) on the LAI values observed in January in *P. massoniana - L. glaber*, *C. axillaris* and *L. glaber - C. glauca* forests.

[Figure]

**Fig. 2.**

**Fig. 4** Partial effects of stem number and the proportion of deciduous species to total stand BA (PDSB) (calculated for overstory trees with height larger than average stand height) on the LAI values observed in July in *P. massoniana - L. glaber, C. axillaris* and *L. glaber - C. glauca* forests.

[Figure]

**Fig. 3.**

---

## Author Response (AR1)

Dr. Akihiko Ito
Associate Editor
Biogeosciences
Email: itoh@nies.go.jp

Dear Editor,

Please find attached the revision of our previous manuscript entitled "Spatial and seasonal variations of leaf area index (LAI) in subtropical secondary forests related to floristic composition and stand characters" (No. bg-2016-5) for publishing as a research paper in the ***Biogeosciences***.

At first, we thank you and the three referees for the constructive and helpful comments on the previous manuscript. We have revised the manuscript accordingly and all changes are highlighted in blue font. You will find our point by point reply to the comments in the following pages. We sincere hope that with these revisions the manuscript is acceptable for publication.

Thank you very much for your consideration. We look forward to hearing from you.

With best regards,

Yours sincerely

Wenhua Xiang

Faculty of Life Science and Technology
Central South University of Forestry and Technology
No. 498 Southern Shaoshan Road
Changsha 410004, Changsha, Hunan
Email: xiangwh2005@163.com

**Responses to comments**

We thank two anonymous referees, Dr. X. Zhang and Associate Editor Dr. Akihiko Ito for their valuable comments on the previous manuscript. We have carefully taken the comments raised by all referees into consideration in revising our paper. Our detailed responses to the comments are presented as follows.

**Editor' comments:**

Thank you for uploading your replies to referee comments and a short comment. I appreciate you responded closely to the helpful short comment. The two anonymous referees evaluated your manuscript positively, such that it is suitable for publication after appropriate revision. Both referees raised several questions on the statistical analyses, which were not adequately described in the manuscript. Therefore, you should carefully revise the manuscript by improving clarity of Materials and Methods. Also, both referees made several recommendations to improve presentation quality of figures and tables.

Re: Thanks for the positive comments and constructive suggestions! We have carefully revised the manuscript according to the comments. As for the statistical analyses, we used GAMs directly rather than two steps (e.g. LSR and GAMs) to examine the effects of stand characters on LAI variation. We also revised the Materials and Methods section and improved the figures and tables. Please see our responses below. We hope that this revision is acceptable for publication.

**Referee #1' s comments:**

The authors investigated seasonal variation, spatial heterogeneity of LAI and its controlling factors by using spatial statistics and generalized additive models (GAM) based on observed values of three forests in subtropical China. They found that LAI values differed greatly by forest types and seasons and showed strong spatial autocorrelation. Species diversity and stand variables like stand density affected LAI values. The work is new for subtropical forests. This is a well-written manuscript well

suited for biogeoscience. The topic is of general interest to readers in the field of forest ecosystem process. I only have a few questions/comments on model parts.

Re: Thanks for the overall positive and valuable comments on our manuscript. Based on comments, we have revised the manuscript. Please see the detailed responses below.

1. The authors mentioned they used GAM followed by linear step regression (LSR). You may directly use GAM for stepwise regression by MGCV packages in R and not necessary perform LSR since GAM could describe both linear and nonlinear relationship.

Re: It is a good point. Based on this comment, we used GAM model directly instead of fitting the model by two steps (that is LSR and GAM). Because there are two packages ("gam" and "mgcv") developed in R project for GAM and the two packages have the same function, we still used gam package for GAM analysis. Our results (see Table 4, Fig. 3 and Fig. 4) showed that the factors affecting LAI variations differed slightly from the results in the previous manuscript, but the effects were significant. We have revised the manuscript accordingly. We hope that our results are satisfactory for publication.

2. In the methods, you need to report which smooth method you used for GAM.

Re: Based on this comment, we have described the method (smooth spline method with two splines) for GAM (see line 227-228 on page 12).

3. For the results of model fitting, you listed some variables which were not statistically significant (p>0.1). For example, BA (p=0.258), crown width (p=0.327) and crown coverage (p=0.333) in Table S1 for LSR and crown width (p=0.209) and crown coverage (p=0.456) for GAM in Table 2. This will change the conclusion on the variables related with LAI. Although the model is not for prediction, you may lower the significant level. Please carefully check the results.

Re: Based on the comments above, we re-run the gam package directly and the variables with statistical significance are presented in the Table 4. Thus, the variables that are not significant are not shown. And we have revised the results and conclusions accordingly in this revision (see Table 4 and the Results and Conclusion sections).

4. Page 11 Line 11. "Tree species diversity" is better than "species diversity".

Re: Changed as suggested (see line 222 on page 12).

5. Page 21 Lines 395-396. It is interesting the authors recommended 30m as a reference for sampling plot size to estimate LAI in subtropical forests. However, you may use a range not point value to account this according to table 3.

Re: Good point! We have replaced the point value with the range value (i.e. from 13m to 27m) based on this comment (see line 431-432 on page 23).

6. As the author mentioned, there are many factors affecting LAI. As an important stand structure characteristic, stand structural diversity (tree size diversity in this case) may explain LAI variation partially. I suggest testing the factor in the study.

Re: Good suggestion! We calculated the tree size diversity based on the reference (Lei XD, Wang WF, Peng CH. 2008. Relationships between stand growth and structural diversity in spruce-dominated forests in New Brunswick, Canada. Can. J. For. Res., 39, 1835-1847). Then we have added this variable to run GAM model (see lines 152-155 on pages 8-9). However, we found no significant effect of tree size diversity on LAI, so did not present the result.

7. Fig. 1. *P. massoniana-L. glaber* and *C. axillaris* cannot be recognized clearly. Please change the legend.

Re: Good point! We have changed as suggested (see Fig. 1).

8. Fig.3. I am wondering you may have wrong values for BA (range from 0 to 6000?) and crown coverage (range from 0 to 1000?). What is the unit for them? Same as Fig. 4. Please carefully check them.

Re: In the previous manuscript the unit for BA was $cm^2$ and for crown coverage was $m^2$. We only used the data of individual trees with height larger than average height in each stand, so some values of BA and crown coverage were within the range. After re-running

the GAM model, only total crown coverage of the stand is a significant variable. We have checked the data carefully and presented the right results (see Fig. 3).

9. Table S1. The summary of values of stem density, BA and IV by species are not equal to the whole stand.

Re: Yes, you are right. In the previous manuscript, the data in Table S1 were for the all species and the top five tree species. The data of other species were not provided in Table S1. Sorry for our carelessness. We have added one row to show the summed data for the rest species (see Table S1).

10. Table S2. Parts of the columns of mean sq and sum aq are the same? Actually you need not to report these values besides parameters, F values and p values.

Re: We have deleted the Table S2 because we used GAM model directly instead of fitting the model by two steps (that is LSR and GAM) based on your comment.

**Referee #2' s comments:**

GENERAL COMMENT The topic of the manuscript lies within the field of the journal Biogeosciences. It reports on spatial and temporal variability of the leaf area index in forests. The overall importance of reliable LAI measurements is undoubted and systematic studies of spatial variability within forest stands are seldom. In this sense, the present study is justified. Unfortunately, the description of the methods is insufficient and the obtained results remain therefore questionable.

Re: Thanks for the positive comments and suggestions on our manuscript. Based on the comments, we have revised the methods and results sections. The detailed responses are presented as follows.

DETAILED COMMENTS Material and methods Since this determines the canopy structure, it should be stated if the studied forest plots where planted or if they are from natural regeneration, further if they were thinned or selectively cut at some point in time.

According to the supplementary table S1, it appears that the stands are unevenaged, but a clear information about their history would be useful.

Re: Good point! The forests are originated from natural regeneration after human disturbance was prohibited in the middle 1960s and no thinning or selectively cutting were applied there till to investigation. The history of the forests was described in the Materials and Methods (see line 125-132 on page 7).

The material used for the hemispherical photography is only poorly described. The camera type is given, but not its manufacturer. There is no information about the lens, not even its viewing angle (or focal length). The choice of picture exposure is not described although it is essential to achieve a good contrast without overexposure. The resolution of the pictures is not given, nor their format.

Re: Thanks for valuable suggestion. We have provided clear information about the material of the hemispherical photography, such as the manufacturer (Shiya Scientific and Technical Cooperation, China), the lens (Pentax TS2V114E, Japan), the viewing angle (180°), the picture exposure (automatic exposure set by the manufacturer), the picture resolution and format (768×494 pix, BMP) (see line 159-164 on page 9).

The picture analysis is also insufficiently described. There is no indication of the software used, of the pixel classification (thresholding), of the considered viewing angle and if it was divided into rings. The viewing angle would be very important to know here because, in conjunction with the tree height, it determines the integration area of the LAI measurement (which is, in turn, important for understanding the spatial variability).

Re: We have revised the manuscript by adding the description of the picture analysis such as the software (the plant canopy analysis software developed by the manufacturer), the pixel classification (thresholding) (752(H)×582(V)), the considered viewing angle (150°) and it was divided into 5 rings (see line 165-168 on page 9).

The method to estimate a clumping factor does not state the number of sectors used. The estimation of the contribution of leaves versus wood to the plant area index would be a

positive aspect of this study, but here also the methods are poorly described. It is not stated if all of the woody elements on all pictures were painted or only sub-samples. Further, "replace the woody materials with surrounding of non-woody materials" is either a wrong wording or a wrong method. Woody areas should neither be replaced by "non-woody materials" nor by sky pixels, they should be excluded from the analysis because it is essentially not known how much leaf area or sky area they hide.

Re: Sorry for our unclear description. We originally described the method according to the reference (Liu ZL, Jin GZ, Chen JM, Qi YJ. 2015. Evaluating optical measurements of leaf area index against litter collection in a mixed broadleaved-Korean pine forest in China. Trees, 29: 59-73), where the word "replace" used means "exclude the pixels of woody materials". We have changed the sentence into "In Photoshop software, we used the Clone Stamp Tool to select the image of the woody materials (e.g., stems) and excluded the pixels, leaving only leaves on the photos" (see line 172-173 and 176-178 on page 10).

Statistical tests are partly done after different types of data transformation. I'm not sure if cutting outliers back to "normally maximal values" is an appropriate method, but at least the measure of this transformation in table 1 should be described in an understandable manner. Using non-parametric statistics would probably make the tests more convincing than the different transformations applied here.

Re: Yes, you are right. Our description is not clear, so we have changed the sentences into "According to Chiang et al. (2003), we regarded the LAI values as the normal values when the LAI values were within mean value±3×standard deviation. Otherwise, the LAI values were outliers and replaced with the maximum or the minimum of normal values. Because the geostatistics analysis requires that the data meet normal distribution, the transformation was applied if the data did not meet normal distribution". To support our method, we have cited the references (Chiang LH, Pell RJ, Seasholtz MB. 2003. Exploring process data with the use of robust outlier detection algorithms. Journal of Process Control, 13(5): 437-449; Dai FQ, Zhou QG, Lv ZQ, Wang XM, Liu GC. 2014. Spatial prediction of soil organic matter content integrating artificial neural network and ordinary kriging in Tibetan Plateau. Ecological Indicators, 45: 184-194) in the text and

added them in the reference list. We hope this revision is clearer than it was before (see line 192-196 on page 11).

Crown coverage is used as a factor in statistical models, but it is not described what this parameter means and how it was measured. A crown coverage is often derived from hemispherical photographs. Is it also the case here, or is it an independent measurement? This can completely change the interpretation of the obtained statistical relationship.

Re: Sorry for our ambiguous description. The crown coverage was not derived from hemispherical photographs and it was calculated from crown diameter measured for individual trees within a stand (see line 145 on page 8).

The kriging is also insufficiently described in the methods section (it is only in a figure legend that it is given as "ordinary"). The maps produced by this kriging show island structures that probably correspond to the grid of picture taking. If this is true, then it indicates a methodological problem. Either the photographs were systematically taken in some spatial relation to the trees (e.g. on a regular grid in a regularly planted stand). Or the very goal of kriging, i.e. interpolating between discrete measurements, was missed.

Re: According to this comments, we have added description of the Kriging in the methods section. Although the ordinary Kriging has the drawback, it is a commonly used interpolating method in the geostatistics reported by other studies (Elbasiouny H, Abowaly M, Abu_Alkheir A, Gad A. 2014. Spatial variation of soil carbon and nitrogen pools by using ordinary Kriging method in an area of north Nile Delta, Egypt. Catena, 113: 70-78. Dai FQ, Zhou QG, Lv ZQ, Wang XM, Liu GC. 2014. Spatial prediction of soil organic matter content integrating artificial neural network and ordinary kriging in Tibetan Plateau. Ecol. Indic., 45: 184-194) (see line 211-218 on page 12).

Results and discussion The presented results would probably be interesting, but due to the poor description of the methods they are all more of less doubtful.

Re: We have revised the Methods section (see Materials and Methods section) and hope

this revision is satisfactory.

Tables and figures Table 1 and 2 should use the same structure to be comparable. Table 3 should include the sample size, otherwise the column RSS is meaningless. Table 4 gives statistical tests without giving any information on how the different factors affect the dependent variable. Since this is not so easy to put in a table in the case of non-linear relationships, table 4 should make a reference to fig. 3. Figure 1: the two grey tones cannot be distinguished.

Re: We have used the same structure for Table 1 and Table 2, and added the sample size in Table 3 as suggested. We have changed Table 4 and Fig. 3, which showed the effect factors and relationship between LAI and factors, respectively. The two grey bars in Figure 1 have been changed into empty and grey, respectively.

Language The English of the manuscript is well understandable but some sentences are not well structured. At least in one case the wording is inappropriate: "throughout four measurement seasons" would mean at least several measurements in each season (while there is actually one per season).

Re: We asked a native English editor from the Charlesworth Group to improve the language (see http://www.charlesworth-group.com).

**X. Zhang's comments:**

The manuscript entitled "Spatial and seasonal variations of leaf area index (LAI) in subtropical secondary forests related to floristic composition and stand characters" by Zhu et al. is an interesting study on the spatial heterogeneity of LAI and its controlling factors in subtropical forests in China. The paper covers an important issue. The investigation is in-depth and thorough. The results are interesting and fill the gap of LAI measurement in subtropical forests. The paper is well-written and duly illustrated. Publication is therefore recommended with minor revisions suggested as follows:

Re: First of all, we thank X. Zhang very much for the positive comments and valuable

suggestions. Based on the following comments, we have revised the manuscript and our detailed replies are presented below.

1. Line 33-34: insert a word "and" after the geostatistics method.

Re: We have added "and" as suggested (see line 28 on page 2).

2. Line 46: remove the keywords "Deciduous species". In your paper, more than one tree species were investigated and the constituents of forests or tree species richness was one of the controlling factors of LAI values. In other words, "deciduous species" is not a proper substitute for the proportion of deciduous species.

Re: We replaced the keyword with "Geostatistical analysis" as suggested (see line 40 on page 3).

3. Line 51, 55-56 and throughout main text, the reference should be arranged by the published year.

Re: Based on the comments, we have changed all references in the entire manuscript according to a chronological order (see line 45 on page 3 and the others).

4. Line 59: insert a word "as" between "used" and "parameter".

Re: Instead of adding "as", we have changed the sentence into "Leaf area index (LAI), defined as total one-sided leaf area per unit ground surface area (Biudes et al., 2014), is a widely used parameter to: : :." (see line 50-55 on page 3).

5. Line 109: change "stand character" to "stand characters".

Re: Changed as suggested (see line 96 on page 6).

6. Line 133-134: the mean temperature of the study site should be a fixed value, please correct it.

Re: We have changed the mean annual air temperature into "16.5 C°"(see line 118 on page 7).

7. Line 148: check and correct the plot size of *P. massoniana - L. glaber* mixed forests.

Re: We have checked and the plot size is correct because the plot of *P. massoniana - L. glaber* mixed forests is irregular with 90 m × 190 m (see line 135 on page 8).

8. Line 170: please add the manufacturer and country to the LAI measuring instrument (SY-S01A).

Re: Added as suggested (see line 159 on page 9).

9. Line 199-200: coefficient of variation (CV) does not need full name here.

Re: We have used abbreviation CV here (see line 189 on page 10).

10. Line 234-238: the author need to report which smooth method used for GAM in this study.

Re: Good point. We have indicated that the smooth method for GAM is smooth spline (see line 227-228 on page 12).

11. Line 250: it is better to illustrate the version of R software used in this study.

Re: We have added the version of R (R 3.2.1) in the manuscript (see line 238 on page 13).

12. Line 255: consider changing "month" in Table 1 into "measurement seasons". Do the same modifications in other tables and Fig. 1.

Re: Change as suggested (see all Tables and Fig. 1 in this manuscript).

13. Line 265: How did you calculate the mean LAI values? I'm a little confused that why you think it's necessary to report the minimum, maximum and mean values of LAI at the same time. what's the differences or the particular meaning between them?

Re: We calculated average LAI values of 100 plots in each forest at a given measurement season. The minimum and maximum values within a forest at different measurement seasons to examine the variations in LAI (see line 190 on page 10).

14. Line 350: ": : :. but they are not suitable for LAI correction in subtropical forests", why? Is this a conclusion drew by yourself or from other's research?

Re: The previous studies by Liu et al. (2015a) and Liu et al. (2015b) showed that the $\alpha$ values ranged from 0.04±0.01 to 0.69±0.12 and $\Omega_E$ values ranged from 0.88±0.04 to 0.96±0.01. These values were measured in temperate forest in northeastern China and differed from our study ($\alpha$ ranged from 0.04±0.03 to 0.15±0.09 and $\Omega_E$ ranged from 0.84±0.09 to 0.92±0.08). Therefore, we drew the conclusion and revised the sentence (see line 330-333 on page 18).

15. Line 360: change "is" to "was".

Re: Changed as suggested (see line 340 on page 18).

16. Line 695-700: the "RSS" in the first line in Table 3 need to be clarified.

Re: We have offered the full name of RSS (residual sum of squares) (see Table 3).

17. Line 745-750: In Fig.1, the y-axis should change into "mean LAI value", x-axis should change into "Month".

Re: We remained the axis labels (see the reply to comment 12).

18. Fig 3 and 4: These two figures are new and unique, and the results might be interesting. It's a pity that you didn't thoroughly discuss these figures except simply described in Results Line 326-330. I suggest to add some discussion about these two figures in you manuscript.

Re: Good suggestions. We have added some sentences to discuss the results of Fig. 3 and Fig. 4 (see line 385-417 on page 21-22).